# The First Representative of the Roachoid Family Spiloblattinidae (Insecta, Dictyoptera) from the Late Pennsylvanian of the Iberian Peninsula

**DOI:** 10.3390/insects13090828

**Published:** 2022-09-12

**Authors:** André Nel, Artai A. Santos, Antonio Hernández-Orúe, Torsten Wappler, José B. Diez, Enrique Peñalver

**Affiliations:** 1Institut Systématique Evolution Biodiversité (ISYEB), Museum National d’Histoire Naturelle, CNRS, Sorbonne Université, Université des Antilles, EPHE, 57 Rue Cuvier, CP 50, 75005 Paris, France; 2Departamento de Xeociencias Mariñas e Ordenación do Territorio, Facultade de Ciencias do Mar, Universidade de Vigo, 36310 Vigo, Spain; 3Centro de Investigación Mariña, Universidade de Vigo (CIM-UVIGO), 36310 Vigo, Spain; 4Departamento de Geología, Universidad del País Vasco/Euskal Herriko Unibertsitatea (UPV/EHU), Part 644, 48080 Bilbao, Spain; 5Department of Natural History, Hessisches Landesmuseum Darmstadt, Friedensplatz 1, 64283 Darmstadt, Germany; 6Department of Palaeontology, Institute of Geosciences, Rheinische Friedrich-Wilhelms-Universität Bonn, Nussallee 8, 53115 Bonn, Germany; 7Instituto Geológico y Minero de España (IGME), CSIC, C/Cirilo Amorós 42, 46004 Valencia, Spain

**Keywords:** Polyneoptera, Holopandictyoptera, *Sysciophlebia*, taxonomy, roachoid families and genera, stratigraphic markers, Carboniferous, Pennsylvanian, Gzhelian

## Abstract

**Simple Summary:**

The Palaeozoic–Early Mesozoic roachoid family Spiloblattinidae are valuable for understanding the stratigraphy of continental strata, thanks to their diversity of the forewing venation patterns of colouration. Here, the first Iberian representative of these roachoids is described as a new ‘form’ closely related to the Gzhelian–early-middle Asselian ‘zone species’ and ‘forms’. It supports the latest Gzhelian age of the concerned outcrop, obtained through stratigraphy and floral composition. It confirms the value of these insects for stratigraphic purposes.

**Abstract:**

*Sysciophlebia* ‘sp. form Villablino’, the first Iberian representative of the Palaeozoic–Early Mesozoic family Spiloblattinidae, is described and illustrated. Its forewing colour pattern is strongly similar to those of the Gzhelian–early-middle Asselian species *Sysciophlebia euglyptica*, *Sysciophlebia ilfeldensis*, *Sysciophlebia rubida*, and ‘*Sysciophlebia* sp. form KBQ’, supporting the currently proposed Gzhelian age for its type locality. It supports the use of the representatives of the Spiloblattinidae for stratigraphic purposes. The diagnoses and limits of the families Subioblattidae, Phyloblattidae, Compsoblattidae, Spiloblattinidae, and of the spiloblattinid genera are discussed.

## 1. Introduction

The current classification of the Palaeozoic and early Mesozoic ‘roachoids’ (also called Eoblattodea Laurentiaux, 1959, after Li [1], representatives of the stem group of the extant superorder Dictyoptera Leach, 1815 (orders Blattodea (including Isoptera), and Mantodea) is confusing. Phylogenetic analysis has never tested the limits and monophyly of the families as currently defined [2]. Consequently, the polarity of the characters used in their diagnoses is not established. Furthermore, in some cases (e.g., the Spiloblattinidae Handlirsch, 1906), different conceptions are employed and not reciprocally discussed [3,4]. Lastly, the diagnoses of the different families differ in a few characters (see discussion below). As a result, attributing any new fossil to a precise family and genus is quite challenging to achieve. Nevertheless, these roachoids can be of great interest for stratigraphic purposes, as convincingly proposed by Schneider and colleagues in several recent publications (see below). It is especially the case for the species of the family Spiloblattinidae (sensu Schneider [3]), for which the wing colouration patterns can be used to define the species and different stratigraphic ‘zones’ that they characterize [5,6].

Here, we describe the first representative of the family Spiloblattinidae from the Palaeozoic of the Iberian Peninsula. We also discuss the different diagnoses of the possibly related families, of the family Spiloblattinidae itself, and of its included genera. This new fossil would confirm the latest Gzhelian age of the outcrop from where it comes.

## 2. Material and Methods

The fossil was collected by Armando Díaz Romeral (Cuenca Province, Spain) in the Orallo mine pit (Villablino Coalfield, León, Spain) (Figure 1). This area corresponds to the ‘Sphenophyllum angustifolium’ Megafloral Zone (Stephanian B sensu stricto) [7]. This site is Gzhelian in age [8]. In the slabs containing the roachoid wing (part and counterpart), remains of pteridophytes of *Oligocarpia leptophylla* (Bunbury in Ribeiro, 1853) Grauvogel-Stamm and Doubinger, 1975, *Lobatopteris corsinii* Wagner et al., 1985, *Polymorphopteris polymorpha* (Brongniart, 1834) Wagner 1959 ex Knight, 1985, and *Cyperites bicarinatus* Lindley and Hutton, 1832, are present, confirming the Gzhelian age. Moreover, in the same slabs, a non-marine bivalve assemblage is recorded. In addition, other insect remains have been identified in the Villablino Coalfield, such as the Dictyoptera *Compsoblatta ovata* Meunier 1921 (= *Phyloblatta monubilis*) [9], and, recently, remains of Archaeorthoptera [10]. On the other hand, evidence of plant–insect interactions has also been reported in the Villablino Coalfield, including evidence of external feeding and oviposition in ferns [11].

Line drawings were done with a camera lucida SZX-DA attached to an Olympus SZX9 stereomicroscope. The photographs of complete slab surfaces and the plant remains were taken using a digital camera Canon EOS 40D. The photographs of the specimen (part and counterpart) were taken with a Canon EOS 650D digital camera using the software ‘Macrofotografía’, version 1.1.0.5 (IGME-CSIC, Madrid, Spain); the software created composite photographs by integrating sequential images obtained at different focal planes. Details of wing venation were taken using Olympus BX53 transmitted-light compound microscope with an attached 38MP FHD Camera V6. Photography was enhanced in Photoshop CS2 version 9.0 (www.adobe.com) to increase contrast to assist in viewing, and the composite figures were prepared in the same software.

We follow the wing venation pattern of Schubnel et al. [12], and the terminology for wing colouration of Schneider and Werneburg [13] adapted to the wing venation terminology. Abbreviations used in the text and figures are as follows: C = costa; CuA = cubitus anterior; CuP = cubitus posterior; M = media; MA = media anterior; MP = media posterior; PCu = postcubitus; RA = radius anterior; RP = radius posterior; ScP = subcosta posterior.

## 3. Systematic Palaeontology


**Clade Holopandictyoptera Kluge, 2010**



**(= total group of extant Dictyoptera Leach, 1815)**



**Plesiomorphon Eoblattodea Laurentiaux, 1959 (sensu Li, 2019)**



**Family Spiloblattinidae Handlirsch, 1906**



**Genus *Sysciophlebia* Handlirsch, 1906**



***Sysciophlebia* ‘sp. form Villablino’**


(Figure 2, Figure 3 and Figure 4)

**Diagnosis.** Forewing characters only. Forewing of medium size, ca. 22 mm long, reduced darkened zones between main veins and along wing margin (where they are continuous); presence of darkened spots in distal part of area between branches of MP; a hyaline zone between veins RA and RP and in zone covered by basal part of branches of RP.

**Description.** Pattern of colouration well visible in part (Figure 2B and Figure 3), but poorly preserved in counterpart, with darkened spots and bands along all branches of main veins, these not being confluent anywhere; in particular, CuP(+PCu) is bordered by a darkened band on each side (CuP-maculae separated in two parts); wing margin with a thin darkened band (costal-maculae, R-maculae and CuA-maculae reduced); wing apex (M-maculae) with a small spot; a transverse darkened zone crossing branches of MP near wing apex, but not touching wing margin; wing margin with a narrow continuous darkened colouration in ScP- and CuA-zones.

Wing 22.3 mm long, 6.8 mm wide (Figure 2 and Figure 4); area between ScP and C 1.5 mm wide; area between ScP and R/RA 0.5 mm wide in widest part, 0.4 mm wide in narrowest part; area between RA and RP 0.5 mm wide; area between R/RP and M/MA 0.8 mm wide; area between M/MP and CuA 0.8 mm wide; radial and median area of nearly the same width; area of CuA broader than median area; ScP very long, reaching distal third of wing, anteriorly pectinate with at least eight branches, nearly all simple and parallel (Figure 2 and Figure 4); RA and RP well-defined; RA distally forked; RP with two main branches, each forked again; MA and MP separated just distad base of RA, MA forked near wing margin, reaching wing apex; MP with two main branches, each forked again; CuA without anterior branch, posteriorly pectinate, with all branches simple except the forked penultimate one; R, M and CuA weakly sigmoidal; CuP(+PCu) regularly curved; five simple and curved veins in anal area, parallel to CuP(+PCu).

**Material****.** Specimen MGM-814Ha-b (formerly ADR 004a-b) (part and counterpart; a nearly complete forewing, with only basal parts of area between C and ScP and anal area not preserved) (Figure 2), stored at Museo Geominero, Instituto Geológico y Minero de España (IGME), CSIC, Madrid.

**Age and outcrop.** Uppermost Gzhelian ‘Stephanian B’; Villablino outcrop, Orallo mine, León Province, Spain (Figure 1).

## 4. Discussion

The very long distal-most branch of ScP (Figure 4) supports affinities with the families Subioblattidae Schneider, 1983, Phyloblattidae Schneider, 1983, Compsoblattidae Schneider, 1978, and Spiloblattinidae Handlirsch, 1906 [3,14].

The diagnoses proposed by Schneider [3] for these families are ambiguous because several characters supposed to define a family are also present in other(s). Affinities with the Subioblattidae (Triassic) are excluded because these have the vein RP and its branches sigmoidally curved [15]. Belahmira et al. [16] proposed the following diagnosis for *Phyloblatta* Handlirsch, 1906: ‘elongate ellipsoidal forewings of about 10–35 mm in length. Costal field strip-like, up to about 70% of forewing length. Sc pectinate; branches end at the anterior wing margin. R sigmoidal; R branches terminate at anterior wing margin. M sigmoidal, multiple-forked, branches covering an area extending from the wing tip to the transition between wing tip and posterior wing margin. CuA sigmoidal. First CuA twigs arise by branching from CuA stem and are rarely forked; all further branches arise by furcation. CuP(+PCu) regularly bended. Crossvenation (archedictyon) reticulate to anastomosing striate’; and the following diagnosis for the Compsoblattidae (unique genus *Compsoblatta* Schlechtendal in Handlirsch [17]): ‘elongate ellipsoidal forewings of about 20–35 mm in length. Costal field strip-like, up to about 60–70% of forewing length. Sc pectinate, branches about 45° inclined to the apex and ending at the anterior wing margin. R sigmoidal, R branches terminate at anterior wing margin. M sigmoidal, multiple forked, branches covering an area extending from the wing tip to the transition between wing tip and posterior wing margin. CuA strongly sigmoidal. First CuA twigs arise by branching from CuA stem and rarely forked; all further branches arise by furcation. CuP(+PCu) regularly bended. The general venation pattern of *Compsoblatta* is similar to that of *Phyloblatta*, but in contrast to this genus the cross venation in the basal three quarters of the wings consists of seams formed by crossvein bases along the main veins and their branches; the remaining wing surface shows a delicate irregularly reticulated cross venation. In contrast to the spiloblattinids, which show similar crossvein seams, the areas between the main veins are not distinctly broadened, and the venation is generally denser in compsoblattids’. The resulting differences are very few, limited to: (1) ‘CuA strongly sigmoidal’ in compsoblattids vs. ‘sigmoidal’ in the Phyloblattidae; in the new fossil, CuA is not strongly sigmoidal (in Schneider [3], compare pl. 3, Figure 1, to pl. 3, Figure 7); and (2) in the Compsoblattidae, the ‘cross venation in the basal three quarters of the wings consists of seams formed by crossvein bases along the main veins and their branches’, which is not the case in the Phyloblattidae and the new fossil.

Nevertheless, Schneider [3] indicated that the Compsoblattidae have darker carbonic ornamentation along the veins, similar to those of the Spiloblattinidae, which is the case for the new fossil.

Schneider [18], p. 24 and [19], p. 28 proposed the following diagnosis for the Spiloblattinidae (translated from German): ‘wings with broad areas between the main veins with few branches. Costal field ribbon-shaped, R and especially M only far from the base distantly forked or branched, CuA sigmoidally curved. Along the wing edge and the veins, there are carbon-like transverse vein hems. These can merge, so that only small and a few bright areas remain free, especially between the main veins’.

Schneider et al. [20] proposed the following emended diagnosis: ‘Phyloblattid-like wing venation pattern but with a much lower number of veins and with extended fields between the main veins. Subcostal field strip-like. Sc pectinate, branches end inclined at the anterior wing margin. R sigmoidal, often with distinct RA, R branches terminate at anterior wing margin. M sigmoidal, multiple-forked, often with distinct MA; M branches covering an area extending from the wing tip to the transition between the wing tip and posterior wing margin. CuA long sigmoidal. First CuA sigmoidal. CuA twigs arise by branching from CuA stem and are rarely forked, all further branches arise by furcation. CuP regularly bent. Anal field with regularly spaced and bent An veins. Most diagnostic is a fenestrate colour pattern consisting of pale areas of various extent between the main veins and their branches. The pale areas do not display a distinct cross-venation. Cross-venation outside the pale areas consists of cross-vein stumps, forming seams along the veins, and in larger dark fields it consists of anastomosing-striate to irregularly reticulate cross veins’.

Thus, except for the pattern of colouration that can or cannot be preserved depending on the fossil (indeed poorly preserved here on the counterpart of the new one), the unique differences with the Phyloblattidae would be the more distal secondary branching of R and M, and the broader areas between the main veins in their basal parts, much narrower in their distal parts, especially between ScP and R and between M and CuA, after the figures in Schneider and Werneburg [13], Schneider ([3], pl. 3, Figure 1, Figure 2, Figure 3, Figure 4, Figure 5 and Figure 6 and Figure 9 and Figure 10) and Schneider et al. ([21], Figure 3). In the new fossil, these areas are not distinctly broadened in their basal parts, even if broader than along the wing margin. The new fossil shares with the Spiloblattinidae the very distal positions of the short secondary branches of M. Nevertheless, after Ross [22], the number of branches of M alone can greatly vary in the roachoids and the extant Blattodea; thus, this character is also not very accurate.

Vishniakova [4] proposed a revision of the Spiloblattinidae to include three subfamilies and much more taxa than in Schneider’s concept. The family diagnosis sensu Vishniakova is as follows: ‘Tegmina are elliptical, elongate or reniform. Costal area is ribbonlike or expanded at base, more rarely in distal part; occupies at least 1/3 of length of tegmen. sc is S-shaped, pectinate or irregularly dividing, R and M are separate or fused at base. R divides into 1R and RS, or 1R is not separated. M divides into two stems or is pectinate, with branches directed anteriorly or, more rarely back or reduced to a simple vein. MP sometimes is present, usually absent. CuA is slopingly convex or S-shaped, pectinate, with branches directed back, sometimes with separated anterior branch. Anal region takes up about 0.40 times the length of tegmen; anal veins are subparallel to CuP(+PCu), 1A is simple or branched distally’. All characters appear heteroclite with contrary options possible, except for the anal veins subparallel to CuP(+PCu). Moreover, all these characters are shared by other families. Thus, we cannot extract any well-defined character state from this diagnosis. Vishniakova also proposed the following diagnosis for the Spiloblattininae (= Spiloblattinidae sensu Schneider): ‘Tegmina. Costal area ribbonlike or gradually narrowed distally. R arched back or weakly sigmoidal. 1R in most genera separated. Intermediate radiomedial and mediocubital areas fusiformly expanded’. Only the last character that corresponds to the characters of Schneider concerning ‘the broader areas between the main veins in their basal parts and much narrower in their distal parts’, can be considered as more or less diagnostic.

This overview shows that the diagnoses of the three families Phyloblattidae, Compsoblattidae, and Spiloblattinidae are very similar and would hardly justify families’ separations for other insect orders. The number of veins in the anal area seems to be lower in the Spiloblattinidae than in the Phyloblattidae (six to eight in former vs. eight to 12 in the latter), but this would need to be confirmed. It is five or six in the new fossil.

Because of the great similarity between the wing venation of the new fossil with those of the Spiloblattinidae (except for the less widened areas between main veins), we attribute it to this family, awaiting a phylogenetic analysis that would clarify the limits and monophyly of these families.

The Spiloblattinidae sensu Schneider’s point of view currently comprises the genera *Spiloblattina* Scudder, 1885, *Syscioblatta* Handlirsch, 1906, *Kinklidoblatta* Handlirsch, 1906, *Sysciophlebia* Handlirsch, 1906, and *Kinneyblatta* Schneider et al., 2021 [14,20].

Schneider [18,19], Schneider and Werneburg [13], and Schneider et al. [20] proposed the following diagnoses for these genera (translated from German):
-*Spiloblattina*: ‘the carbonaceous seams widen distally and largely merge with one another. The transverse vein seams go into a reticulated. Intermediate veins or the entire wing surface are covered with a charcoal membrane except for a few light areas. The areas always without intermediate veins or barely visibly reticulated, mostly light brownish’ [18]; ‘Band-shaped costal area. Sigmoidal and distally frequently bifurcated CuA. Radial area relatively wide distally, M forked or branched approximately from the middle of the wing. RA and MA mostly clearly developed. Areas expanded, especially that between M and CuA. Intermediate veins with maculae and fasculae. The most important feature is CuP(+PCu) bordered by fasculae (hyaline zone) either on one side or on both sides’ [19]; ‘Costal area band-shaped. CuA sigmoidal, first side vein usually forked one to two times; after two to four additional side veins, CuA forks distally several times. R with mostly clear R1. M with MA area branching off approximately in middle the wing; expanded areas between main veins, especially that between M and CuA; wing surface with bright maculae located in costal area and in form fasculae at each wing tip. This colour pattern is becoming increasingly reduced in younger forms to the point of disappearance’ [13].-*Kinklidoblatta*: ‘less densely veining. Costal area long, palmate. M forked or branched from about the middle of the wing. CuA relatively straight or sigmoidal. Areas between main veins expanded, especially the M-CuA area. Intermediate veins reticulate, in the apical part of wing tip and in the fields often more densely reticulated’ [19].-*Syscioblatta*: ‘costal area band-like; CuA sigmoidal, distally bifurcated. R with relative RA branching off far distally; M from about the middle of the wing forked or branched, MA pronounced. Areas between main veins expanded, especially that between M and CuA. Main and partly side veins with intervein seams, forming maculae and fasculae. These hems are missing on both sides along the CuP(+PCu) at *Syscioblatta*, and a broad band-like CuP-fascula enclosing CuP(+PCu) is formed’. This is the most essential feature to distinguish from the two other very similar genera *Spiloblattina* and *Sysciophlebia* [13].-*Sysciophlebia*: ‘the carbonaceous seams run distally with a constant width. Only between R and M and directly at the wing tip do they merge with each other. Fields hardly reticulated, light brownish’ [18]. ‘Band-shaped costal area. Sigmoidally curved and distally moderately forked CuA. R and especially M only forked or branched far away from their base. MA usually clearly developed. Most essential and at the same time from *Spiloblattina* distinguishing feature is a charcoal border of transverse veins on both sides of the CuP(+PCu), so that the fasculae do not touch the CuP(+PCu)’; ‘As for the family with the following generic characters. CuA framed on both sides by a dark seam of cross-vein stumps. Pale areas between Sc veins distinctly separated, not merging with each other. Distinct sexual dimorphism in wing geometry and colour pattern, with stouter and darker coloured wings in males’ [19,20].-*Kinneyblatta*: ‘*Kinneyblatta* differs from the so far known spiloblattinid genera *Sysciophlebia*, *Spiloblattina*, and *Syscioblatta* … in having a much denser venation pattern because of a higher number of twigs at the main veins. Additionally, it differs from these genera by the colour pattern of the proximal CuA and the CuP, where a large pale area stretched uninterrupted from the first branches of CuA to the CuP’ [20].


Here too, the differences between the genera are few, so their monophyly is not accurate. Nevertheless, affinities of the new fossil with *Kinklidoblatta* can be excluded because the veins R and M are forked distad mid-wing, and the areas between the main veins are not broadly expanded. The CuP-fasculae do not touch the vein CuP(+PCu) on both sides in the new fossil, supporting an attribution to *Sysciophlebia* rather than to *Spiloblattina* or to *Syscioblatta* (see Hmich et al. [23], Figure 3). The new fossil also differs from *Kinneyblatta* in the pattern of coloration and the clearly less numerous branches of RA.

Vishniakova [4] proposed to exclude *Kinklidoblatta* from the Spiloblattinidae, ‘because it is characterized by early development of a specialization contrary to spiloblattinids—a distinctive wing covering’, which remains an unclear argument. She also considered the synonymies proposed by Schneider for a series of Spiloblattininae not well supported. Vishniakova ignored the characters of colour patterns proposed by Schneider to separate the genera, who does not consider the work of Vishniakova in his papers more recent than 1993, see [20], p. 269. This taxonomic problem needs to be revisited, maybe with a different approach. Vishniakova proposed a key to ‘spiloblattinine’ genera. The new fossil would fall in the genus *Sysciophlebia* because of the following characters: ‘M fissile, CuA with 1–2 simple anterior branches or without them’; ‘M pectinate, intermediate radiomedial area narrower than mediocubital’; ‘Branches of M directed forward’; ‘Costal area relatively broad, takes up at least than 0.2 times width of tegmen. R and M equally developed or R richer than M’; ‘Tegmen elliptical, CuA S-shaped, archedictyon preserved along veins, perforated or reticulate’. Additionally, the new fossil does not fit in any of the two other subfamilies of the Spiloblattinidae sensu Vishniakova, viz. Vivablattinae Vršanský and Mostovski, 2000 (replacement name for Kargaliinae Vishniakova, 1993 [24]) or in the Permoblattininae Vishniakova, 1993, because of the very different wing shape and ScP, among other characters.

Among the *Sysciophlebia* spp., the colour pattern of the new fossil fits well with those of *Sysciophlebia euglyptica* (Germar, 1851), *Sysciophlebia hercynica* (Scharf, 1924), *Sysciophlebia praepilata* (Meunier, 1921) (holotype MNHN-F.R51102), *Sysciophlebia ilfeldensis* Schlechtendal, in Handlirsch [17], and *Sysciophlebia rubida* Schneider, 1982 (see [5], Figure 2; [6], Figure 3; [12], pl. 4, Figure 5, Figure 6 and Figure 7; [19], Figure 6), in the reduced darkened zones along the posterior margin and main veins. Other species of Spiloblattinidae (except *Kinklidoblatta*) have distinctly broader dark zones in their wings.

Schneider [19] proposed the following diagnosis for *S. euglyptica* (translated and adapted from German): ‘forewing 3.0 cm × 1.0 cm, with sparse phyloblattoid veining, band-shaped costal area and sigmoidally curved CuA moderately forked distally. R and especially M only forked or branched far away from the base. MA is usually clearly developed. The most important feature, which also differs from *Spiloblattina*, is a carbonaceous transverse vein border on both sides of the Cu2, so the fasculae do not touch the CuP+PCu’. He proposed the following one for *S. hercynica*: ‘forewing 16 mm long and 5 mm wide, markings basically like *S. euglyptica*, but the maculae and fasculae are even more extensive, so that the vein seams are more delicately developed; more branches at CuA than *S. euglyptica*’; the following diagnosis for *S. ilfeldensis*: ‘approximately 3.2 × 1.13 cm large forewing. Contour fascula separate, only in the distal M area fused maculae, which only run there and in the distal CuA area up to the vicinity of the alar edge, going back proximally in the CuA area continuously to less than 1/2 vein length; anal fascula and maculae shortened; costal fascula tapered proximally’; for *S. praepilata*: ‘forewing 3 cm long, contour fascula consisting of separate maculae throughout (distal M area not preserved). In the area of the last R and first M branches in the distal half of the wing, the maculae are larger than in *S. rubida*. Between the CuA branches they reach closer to the posterior margin. In contrast to the nearby *S. euglyptica*, the black intervein border accompanying the posterior edge is even broader on the first CuA branches’; and the following diagnosis for *S. rubida*: ‘forewing ca. 2.5 × 0.9 cm, contour fascula without gap. However, separate maculae fused only in the distal part of median region; there the maculae also reach the margin of the wing, otherwise, the other maculae are a certain distance from wing margin, which is largest at the wing tip and in the area of the first CuA branches’.

Thus, the main differences between these species concern the patterns of forewing colouration and the number and arrangement of the branches of the main veins.

The intraspecific stability and specific value of the pattern of forewing colouration among the extant Blattodea remain poorly known, and are generally not used in the species descriptions and diagnoses. Hebard [25], p. 97, characterized (in part) the species of the ectobiid genera *Ellipsidion* Saussure, 1863 and *Balta* Tepper, 1893 after the different patterns of wing colouration; however, in the extant species *Ellipsidion gemmiculum* or *Balta stylata*, males and females can have different patterns of black colouration, even if globally similar to those of the Spiloblattinidae, after Rentz [26], pp. 593, 608. Moreover, *Balta hebardi* and an ‘undescribed species related to *Balta hebardi*’ have extremely similar patterns of forewing colouration, and similar to those of the Spiloblattinidae, after Rentz [26], pp. 20, 579–580. Thus, the specific value of the different patterns of colouration of the forewings in the Spiloblattinidae remains questionable.

Schneider et al. [20], p. 271, indicated that ‘the *Sysciophlebia* lineage (as in those of *Syscioblatta* and *Spiloblattina*, too) represents an anagenetic change of features, i.e., it is one chronospecies only, which is subdivided for practical biostratigraphic use into zone species. Because of the flowing change of features through time and to prevent the implementation of too many useless “new” species, we use for transitional forms the designation “form” (instead of subspecies) with a letter or combination of letters that are derived from the finding locality.’

This sentence would suggest that all the species in *Sysciophlebia* would be ‘forms’ of one species. The problem is that Schneider et al. [20] did not make any synonymy between the ‘zone species’ that were previously described. The resulting situation is very confusing.

The new fossil differs from *Sysciophlebia praepilata* in the smaller forewing (22 mm long vs. 30 mm in the latter), and continuous colouration along the costal margin vs. disposed in separated maculae. The new fossil and *Sysciophlebia ilfeldensis* differ from *Sysciophlebia euglyptica* and *Sysciophlebia hercynica* in the presence of darkened spots in the distal part of the area between the branches of MP. The new fossil is also smaller than *S. euglyptica* and larger than *S. hercynica*. The new fossil differs from *Sysciophlebia ilfeldensis* in its smaller size (forewing ca. 22 mm long vs. 32 mm in the latter), in the hyaline zone between the veins RA and RP, and the hyaline zone between the branches of RP in their basal part. It would better fit with *Sysciophlebia rubida* (Kasimovian to Gzhelian, Stephanian B) and the Kasimovian ‘*Sysciophlebia* sp. form KBQ’ (Kinney Brick Quarry) in the forewing length (ca. 20 to 24 mm long) and pattern of coloration (especially with the specimen NMMNH P-77159a) [20,21].

Thus, we are unable to create a new species that would be justified by the wing size and pattern of colouration of the forewing, given the very confusing state of art. We attribute it to a new form of *Sysciophlebia* ‘sp. form Villablino’, closely resembling ‘*Sysciophlebia* sp. form KBQ’.

## 5. Palaeoecological Comments

Patterns of forewing coloration similar to those of the Sysciophlebia spp. are found in some extant species of the cockroach genus *Balta*. These live in tree trunks, bark, ground and plant remains [26]. The exact function of these wing patterns, possibly disruptive, remain unknown.

During the Gzhelian, the Cantabrian land basins were in the foreland of the Hercynian mountain range, close to the Palaeotethys Sea [27,28]. Ecosystems were developed at low altitudes [29,30,31], in positions close to the equator, less than 5° south [32]. During the Late Carboniferous, the climate was tropical and humid (although maybe with a dry station), with a tendency towards aridification with the passage to the Permian [33].

The fossil wing was found in a plane of the slab with an assemblage of non-marine bivalves and some remains of *Cyperites*, which are the leaves of the generally hygrophilous *Sigillaria*. On the other slab surfaces there were diverse remains of plants, mainly pteridophytes, including *Oligocarpia leptophylla*, *Lobatopteris corsinii*, *Polymorphopteris polymorpha* (Figure 5), and *Cyperites bicarinatus*. These species may be associated either with communities of tree ferns or mixed communities of tree ferns and pteridosperms. These plant assemblages generally developed in basal areas, on moist mineral soils, or ephemeral swamps, far enough from rivers to avoid being disturbed by sediment discharges [34,35,36].

Therefore, we can infer that this roachoid lived in a mosaic of moist but well-drained mineral soils with freshwater ponds or lakes, and that the concrete wing found fell into one of these ponds or lakes.

This new first representative of the Spiloblattinidae family increases the known diversity of the entomofauna from the Spanish Late Carboniferous coalfields of León, where different orders of insects were previously identified (e.g., Paoliida, Palaeodictyoptera, Megasecoptera, Archaeorthoptera, or Dictyoptera) [10]. These insects cover a wide range of ecological niches (herbivory, carnivory, detritivory). In addition, the plant remains found in the assemblage with the new roachoid are consistent with the previous interpretation of the Pennsylvanian forests of this area, which would be fern-dominated communities in which different plant–insect interactions were found, participating in a palaeoenvironment increasingly rich in known details [11,37].

## 6. Conclusions

The genus *Sysciophlebia* is currently known from the Permian of Austria, the Czech Republic, Germany, South Africa, and the United States, and the Carboniferous of the Czech Republic, France, Germany, Morocco, and the United States (Fossilworks database). Thus, *Sysciophlebia* ‘sp. form Villablino’ represents the first record of the genus from the Iberian Peninsula, and also the first evidence of the Spinoblattinidae in the area.

The ‘Sysciophlebia praepilata’ insect zone is attributed to the ‘Stephanian B/C’ [38] (latest Gzhelian to earliest Asselian, 303.4 to 298.9 Ma, Commentry, France). The ‘Sysciophlebia euglyptica’ insect zone is attributed to the ‘Stephanian C’ (latest Gzhelian to earliest Asselian, 303.4 to 301.2 Ma, Ottweiler Stufe Formation, Germany), while the ‘Sysciophlebia ilfeldensis’ and ‘Sysciophlebia hercynica’ insect zones are slightly younger (early–middle Asselian, 298.9 to 295.0 Ma, Ilfeld Basin, Germany) [6,19,39]. Schneider et al. [20] indicated that the Kasimovian ‘*Sysciophlebia* sp. form KBQ’ could ‘represent a regional North American variation or subspecies of *Sysciophlebia rubida* from the Kladno basin, Czech Republic, Slaný Formation, Hředle Member; Stephanian B (Kasimovian to Gzhelian).’ All these localities belonged to the Euramerican continent during the Late Carboniferous, and to ‘tropical everwet forests or subtropical forests, sometimes seasonally dry’ [40].

Thus, the new fossil would correspond to a Gzhelian–early Asselian age, which fits well with the previous interpretations [7,10,41] based on the flora, attributing an age of late Gzhelian to the outcrop.

## Figures and Tables

**Figure 1 insects-13-00828-f001:**
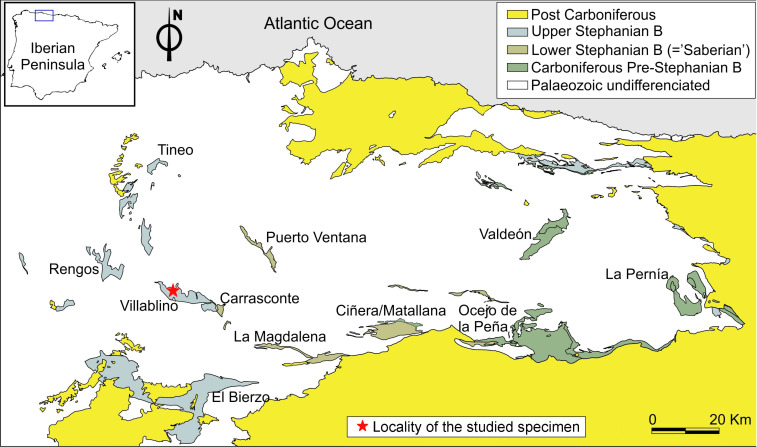
Distribution of continental and transition deposits corresponding to upper Moscovian (Asturian) to lower Permian interval of NW of the Iberian Peninsula of the new specimen. Modified from Santos et al. [11].

**Figure 2 insects-13-00828-f002:**
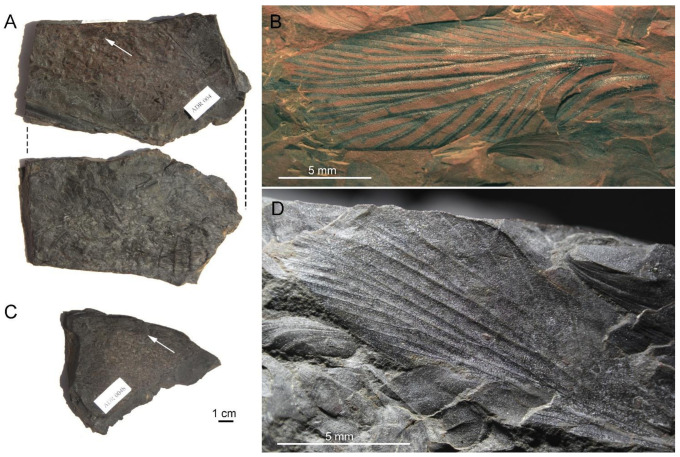
Specimen of *Sysciophlebia* ‘sp. form Villablino’ (Spiloblattinidae), forewing from Orallo mine (Villablino, León Province). (**A**) Two sides of slab containing part, MGM-814Ha (white arrow). (**B**) Wing part, under alcohol. (**C**) Slab with counterpart, MGM-814Hb (white arrow). (**D**) Wing counterpart. Note the abundant bivalve shells on sides A, having the part and counterpart of the fossil wing, and abundant plant remains on reverse. Images (**A**,**C**) to the same scale.

**Figure 3 insects-13-00828-f003:**
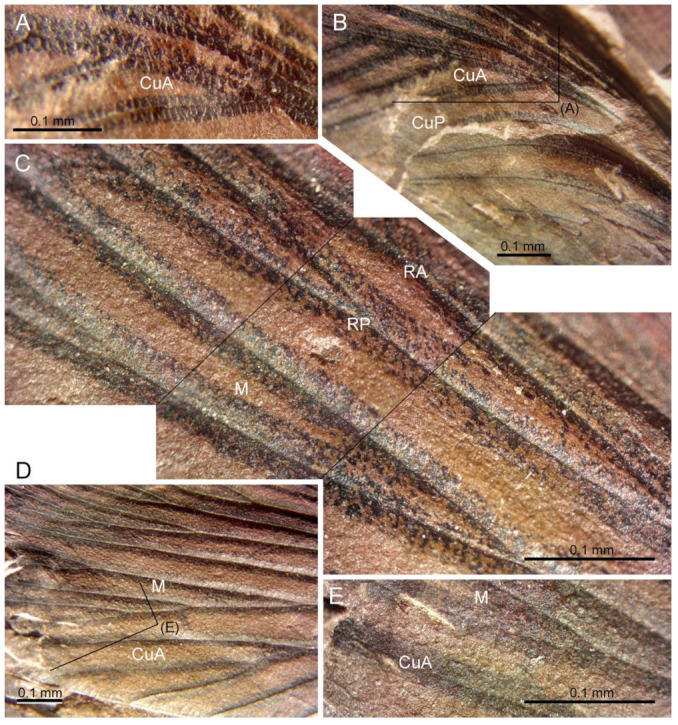
*Sysciophlebia* ‘sp. form Villablino’ (Spiloblattinidae), MGM-814Ha (part), Orallo mine (Villablino, León Province), details of forewing venation and colour pattern. (**A**) First branches of CuA. (**B**) Mid-basal fourth of wing, rectangle corresponding to (**A**). (**C**) R with base of RA and basal branches of M. (**D**) Distal parts of M and CuA, rectangle corresponding to (**E**). (**E**) Apical part of CuA and M. Image (**C**) composed of three photographs. Abbreviations: CuA = cubitus anterior; CuP = cubitus posterior; M = median; RA = radius anterior; RP = radius posterior; ScP = subcosta posterior.

**Figure 4 insects-13-00828-f004:**
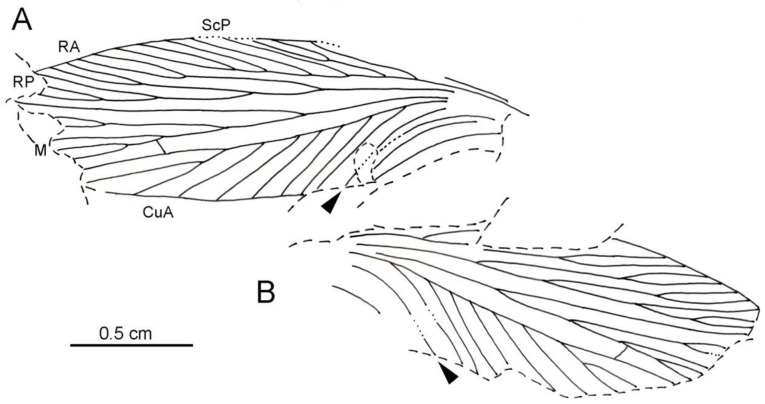
*Sysciophlebia* ‘sp. form Villablino’ (Spiloblattinidae), MGM-814H, Orallo mine (Villablino, León Province), forewing venation, camera lucida drawings. (**A**) Part MGM-814Ha. (**B**) Counterpart MGM-814Hb. Abbreviations: CuA = cubitus anterior; M = media; RA = radius anterior; RP = radius posterior; ScP = subcosta posterior; arrowheads CuP+PCu cubitus posterior + postcubitus.

**Figure 5 insects-13-00828-f005:**
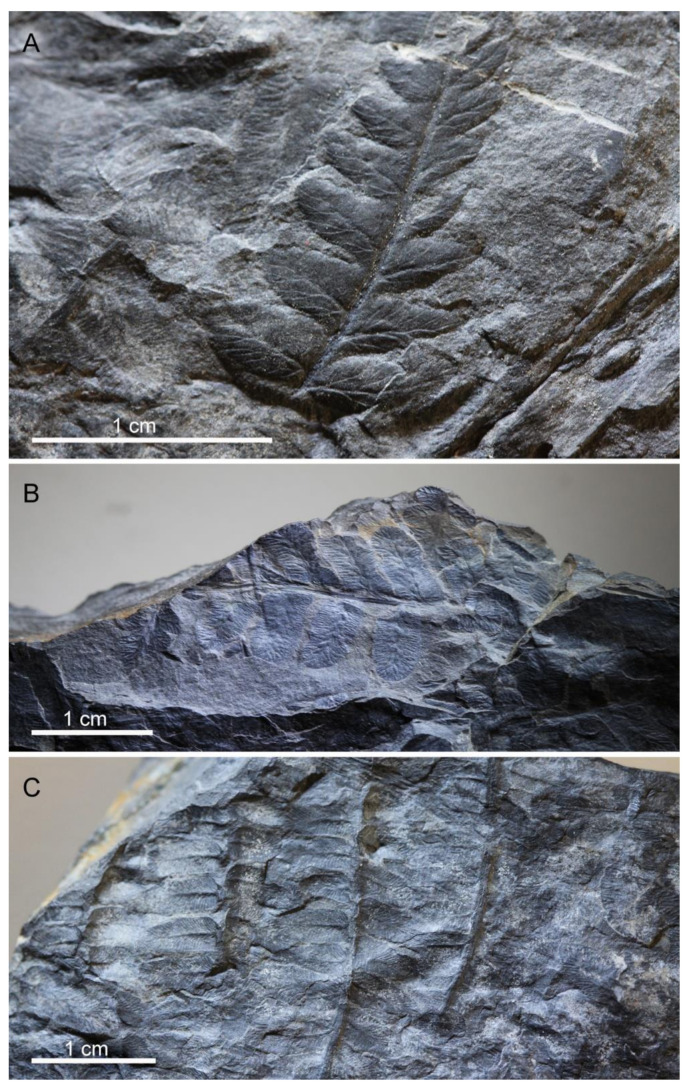
Plant remains recorded on side B of the slab containing the specimen (part) of the roachoid wing. (**A**) *Oligocarpia leptophylla*. (**B**) *Lobatopteris corsinii*. (**C**) *Polymorphopteris polymorpha*.

## Data Availability

The data presented in this study are available in the article.

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
