# Peer review of "The First Representative of the Roachoid Family Spiloblattinidae (Insecta, Dictyoptera) from the Late Pennsylvanian of the Iberian Peninsula"

_insects, 2022, doi:10.3390/insects13090828_

Round 1
Reviewer 1 Report
In this study, a common problem existing in the taxonomy of palaeoentomology is involved. Some characters usually used for identify extinct insects are not used in extant species, the taxonomic value of those characters is not clear. Therefore, the work of this paper, and the taxonomic treatment of new fossil materials are very meaningful and valuable
1. the figure 3. A and B showing the details of cubitus looks not horizontal
2. in line 344, What does the word “disposal” mean here? Do you mean “arrangement”?
3. the abbreviation KBQ’ maybe needs to be specified when it first appears.
4. Have the authors considered using geometric morphological methods to analyze these “species” of Sysciophlebia?
5. One question is that coloration does be specific to some extent, but fossils are sometimes affected by preservation, and sometimes misjudgment if coloration is considered as important or the only distinguishing character.
Author Response
Please, find our answers to the referees
We agree with all comments
Except for the coloring of the drawings, we have preferred to replace one figure by a better one in which the color pattern is well visible
The changes are indicated in red in the text
Thanks a lot for your kind help
Andre nel
Open Review
( ) I would not like to sign my review report
(x) I would like to sign my review report
English language and style
( ) Extensive editing of English language and style required
( ) Moderate English changes required
( ) English language and style are fine/minor spell check required
(x) I don't feel qualified to judge about the English language and style
Yes |
Can be improved |
Must be improved |
Not applicable |
|
Does the introduction provide sufficient background and include all relevant references? |
( ) |
(x) |
( ) |
( ) |
Are all the cited references relevant to the research? |
(x) |
( ) |
( ) |
( ) |
Is the research design appropriate? |
(x) |
( ) |
( ) |
( ) |
Are the methods adequately described? |
(x) |
( ) |
( ) |
( ) |
Are the results clearly presented? |
( ) |
(x) |
( ) |
( ) |
Are the conclusions supported by the results? |
(x) |
( ) |
( ) |
( ) |
Comments and Suggestions for Authors
In this study, a common problem existing in the taxonomy of palaeoentomology is involved. Some characters usually used for identify extinct insects are not used in extant species, the taxonomic value of those characters is not clear. Therefore, the work of this paper, and the taxonomic treatment of new fossil materials are very meaningful and valuable
the figure 3. A and B showing the details of cubitus looks not horizontal
it is not very important, the veins itself is not horizontal
- in line 344, What does the word “disposal” mean here? Do you mean “arrangement”?
Changed thanks
- the abbreviation KBQ’ maybe needs to be specified when it first appears.
added
- Have the authors considered using geometric morphological methods to analyze these “species” of Sysciophlebia?
Yes, but it is out of the scope of the paper, because it means to analyse all the available wings,
- One question is that coloration does be specific to some extent, but fossils are sometimes affected by preservation, and sometimes misjudgment if coloration is considered as important or the only distinguishing character.
Yes, we agree, we have noticed even that for the new fossil, the counterpart does not show clearly the coloration
And we have also indicated that in extant taxa, similar colors are not specific at all
Reviewer 2 Report
I appreciate the presence of the diagnoses and their translations from German since the original papers are not available online. However, it makes the Manuscript hard to read without making extra notes.
My notes and suggestions are included in the pdf file below.
I suggest accepting the Manuscript for publishing, with minor revisions.

Author Response
Review Report Form
Open Review
( ) I would not like to sign my review report
(x) I would like to sign my review report
English language and style
( ) Extensive editing of English language and style required
( ) Moderate English changes required
( ) English language and style are fine/minor spell check required
(x) I don't feel qualified to judge about the English language and style
Yes |
Can be improved |
Must be improved |
Not applicable |
|
Does the introduction provide sufficient background and include all relevant references? |
(x) |
( ) |
( ) |
( ) |
Are all the cited references relevant to the research? |
(x) |
( ) |
( ) |
( ) |
Is the research design appropriate? |
(x) |
( ) |
( ) |
( ) |
Are the methods adequately described? |
(x) |
( ) |
( ) |
( ) |
Are the results clearly presented? |
(x) |
( ) |
( ) |
( ) |
Are the conclusions supported by the results? |
(x) |
( ) |
( ) |
( ) |
Comments and Suggestions for Authors
I appreciate the presence of the diagnoses and their translations from German since the original papers are not available online. However, it makes the Manuscript hard to read without making extra notes.
My notes and suggestions are included in the pdf file below.
I suggest accepting the Manuscript for publishing, with minor revisions.
Adding the colouration would be appreciated by the readers. It is not seen very well on the photo from Fig 2.
We have chosen to change one photograph into a color one, in which the pattern is well visible from the locality where it was found.
It is a citation, we cannot change it, sorry.
A short mention on the reason of the colouration in Spiloblattinid roaches would be interesting to see.
It is still unknown in extant taxa with similar patterns, but we have added an sentence
It would be very interesting to see your interpretation on the ecological niche of the new species, compared with the niches of the insects you mention here
Added: These insects cover a wide range of ecological niches (herbivory, carnivory, detritivory).
In order to save resources, consider removing the photographs of plant fossils.
It is the choice of the editor, indeed
The reasoning behind the assignment to the genus Sysciophlebia would be beneficial, especially if you paid so much attention to it in the Discussion.
Well, here, it is not the place to go again into the taxonomic discussion, we have made it extensively before, no?
A mention of paleogeography would be appreciated. How were these localities related back in Carboniferous?
Yes, added, with a reference, thanks